# The 2015-2016 El Niño increased infection parameters of copepods on Eastern Tropical Pacific dolphinfish populations

Ana María Santana-Piñeros[1,2,3]*, Yanis Cruz-Quintana[1,2,3], Ana Luisa May-Tec[4], Geormery Mera-Loor[2,5], María Leopoldina Aguirre-Macedo[4], Eduardo Suárez-Morales[3], David González-Solís[3]

**1** Grupo de Investigación en Sanidad Acuícola, Inocuidad y Salud Ambiental, Escuela de Acuicultura y Pesquería, Universidad Técnica de Manabí, Bahía de Caráquez, Manabí, Ecuador, **2** Departamento Central de Investigación, Universidad Laica "Eloy Alfaro" de Manabí, Manta, Manabí, Ecuador, **3** El Colegio de la Frontera Sur (ECOSUR), Unidad Chetumal, Chetumal, Quintana Roo, México, **4** Laboratorio de Patología Acuática y Parasitología, Departamento de Recursos del Mar, CINVESTAV-IPN Unidad Mérida, Mérida, Yucatán, México, **5** Carrera de Tecnología Superior en Acuicultura, Instituto Tecnológico Superior "Luis Arboleda Martínez", Secretaría de Educación Superior, Ciencia, Tecnología e Innovación, Jaramijó, Manabí, Ecuador

\* anasantana4@gmail.com

**Data Availability Statement:** All relevant data are within the paper and its Supporting Information files

## Abstract

The oceanographic conditions of the Pacific Ocean are largely modified by El Niño (EN), affecting several ecological processes. Parasites and other marine organisms respond to environmental variation, but the influence of the EN cycle on the seasonal variation of parasitic copepods has not been yet evaluated. We analysed the relation between infection parameters (prevalence and mean intensity) of the widespread parasitic copepods *Caligus bonito* and *Charopinopsis quaternia* in the dolphinfish *Coryphaena hippurus* and oceanography during the strong 2015–16 EN. Fish were collected from capture fisheries on the Ecuadorian coast (Tropical Eastern Pacific) over a 2-year period. Variations of sea surface temperature (SST), salinity, chlorophyll a (Chl-*a*), Oceanic Niño Index (ONI), total host length (TL) and monthly infection parameters of both copepod species were analysed using time series and cross-correlations. We used the generalised additive models for determine the relationship between environmental variables and infection parameters. The total body length of the ovigerous females and the length of the eggs of *C. bonito* were measured in both periods. Infection parameters of both *C. bonito* and *Ch. quaternia* showed seasonal and annual patterns associated with the variation of environmental variables examined (SST, salinity, Chl-*a* and ONI 1+2). Infection parameters of both copepod species were significantly correlated with ONI 1+2, SST, TL and Chl-*a* throughout the GAMLSS model, and the explained deviance contribution ranged from 16%-36%. Our results suggest than an anomaly higher than +0.5˚C triggers a risen in infection parameters of both parasitic copepods. This risen could be related to increases in egg length, female numbers and the total length of the ovigerous females in EN period. This study provides the first evidence showing that tropical parasitic copepods are sensitive to the influence of EN event, especially from SST variations. The observed behaviour of parasitic copepods likely affects the host populations and structure of the marine ecosystem at different scales.

**Funding:** The work was supported by the Departamento Central de Investigación of Universidad Laica Eloy Alfaro de Manabí (CUP 91740000.0000.377806) and the Universidad Técnica de Manabí (Biodiversidad de parásitos metazoarios en peces desembarcados en los cantones Manta y Sucre, provincial de Manabí, Ecuador). The funders had no role in study design, data collection and analysis, decision to publish, or preparation of the manuscript.

**Competing interests:** The authors have declared that no competing interests exist

## Introduction

The El Niño-Southern Oscillation (ENSO) creates fluctuations in the sea surface temperature (SST) of the Tropical Eastern Pacific (TEP), and it has been suggested that its frequency and magnitude will increase in the future [1,2]. During El Niño (EN) events, the equatorial surface waters become considerably warmer (+0.5˚C), particularly along the western coast of South America [3], which has a profound influence not only on the tropical Pacific, but also on many regions all over the world [4,5,6,7,8].

Parasites of aquatic organisms are sensitive to changes in abiotic factors, such as temperature [9,10,11], precipitation [12,13], salinity [14,15], hurricanes [16], the North Atlantic Oscillation (NAO) [8], EN [6] and ENSO [8]. Several studies have shown that climate change strongly affects performance, population dynamics and distribution of marine organisms [6,17,18,19]. In particular, ENSO affects host-pathogen relationships of some marine organisms by reducing host immunity and increasing pathogen virulence [8,16,19,20,21,22]. However, most of these studies have been carried out on fungal, viral, protozoan [8,22,23,24,25] or internal metazoan parasites [9,26,27], while ectoparasites, which are more exposed and influenced by environmental variables, remain poorly studied.

One of the most significant oceanographic features of the upper-ocean circulation system off the Ecuadorian coast is the Equatorial Front. This system isolates the cold, nutrient-rich waters of the Humboldt Current moving northwest and the South Equatorial Current from the warmer, nutrient-poor surface waters in the north [28]. This oceanographic pattern changes under the influence of EN and is characterised by unusually warm SSTs in the equatorial Pacific [3]. Along the coast of South America, EN weakens the upwelling of cold, nutrient-rich water, thus affecting the abundance, composition and distribution of phytoplankton, zooplankton and pelagic fish communities [29,30].

Parasitic copepods affect the growth, fecundity and survival of both wild and farmed fish hosts [31,32,33]. These parasites attach to the skin or gills of their hosts to feed on their mucus, tissues or blood and leave open wounds that can result in secondary infections, leading to increased mortality at high infection levels [33,34]. Two of the parasitic copepods infecting the dolphinfish *Coryphaena hippurus* Linnaeus, 1758 in the Atlantic Ocean and the Mediterranean Sea are *Caligus bonito* Wilson, 1905 and *Charopinopsis quaternia* Wilson, 1935 [35,36,37]. The dolphinfish is an oceanic epipelagic fish found worldwide in tropical and subtropical waters, where it is sought after by commercial and recreational fishers [38]. *Caligus bonito*, a species of the family Caligidae, has also been reported on several other fish hosts (i.e., *Auxis rochei* (Risso, 1810), *Katsuwonus pelamis* (Linnaeus, 1758), *M. curema* Valenciennes, 1836, *Mugil liza* Günther, 1880 reported as *M. platanus*, among others) [39,40], while *Ch. quaternia*, a species of the family Lernaeopodidae, is a little known species, reported mainly from dolphinfish [35]. Members of the family Caligidae are responsible for most of the documented disease outbreaks in cultured marine species [32]. Parasitism in mariculture and wild populations could be better managed by knowing the responses of the parasitic copepods to environmental changes in the eastern tropical Pacific Ocean.

In this context, the goal of this study was to determine the correlation between the infection parameters of *C. bonito* and *Ch. quaternia* in the dolphinfish *C. hippurus* during the non-EN and EN periods of two consecutive years (2015 and 2016).

## Materials and methods

### Collection of fish hosts

Fish were collected from the fishing port "Playita Mía" Beach (00˚57′0.92″S; 80˚42′29.47″W), in the city of Manta, Manabí, Ecuador. This port holds the most extensive records of

dolphinfish landings for the period 2008–2012, among all Ecuadorian fishing ports [28] (Subsecretaría de Recursos Pesqueros, Viceministerio de Acuacultura y Pesca, unpubl. data). Ecuador has two distinct seasons: the wet/warm (rainy) season from December to April and the dry/cold (dry) season from June to November [41], with the air temperature ranging from 26˚C during the dry season to 31˚C during rainy season [42].

The Ecuadorian artisanal fishing fleet for large pelagic species is divided into the inshore and the oceanic fleet, depending on the operational distance from the coast [28]. The inshore fleet is formed by small-size fibreglass boats fishing for 2–3 days in "inshore" waters, 40–200 nautical miles (M) from the coastline, while the oceanic fleet consists of medium to large size "mother-ship" boats, fishing for up to 25 days and reaching the 100˚W longitude beyond the Galapagos Archipelago and as far west as 94˚W south of the Peruvian coast [28]. The dolphinfish used in this study were captured by the inshore fleet and were therefore relatively fresh and not excessively manipulated.

The dolphinfish were caught by using long lines over 2 years, in 1-month intervals (December 2013 to November 2015). The non-EN sampling period was from December 2013 to February 2015, whereas the EN period was from March to November 2015. Due to the absence of previous reports on the prevalence of both parasite species from the TEP, we used a preliminary sampling of 154 dolphinfish to establishing baseline prevalence during the non-EN period (81% for *C. bonito* and 31% for *Ch. quaternia*). The accepted levels of risk ($\alpha = 0.05$) and sensitivity of the sampling and identification methods for the parasites were considered. We assumed that such diagnostic methods have a sensitivity of 75% due to human error (through handling and identification). Thus, assuming a Poisson distribution for the probability of identifying *C. bonito* and *Ch. quaternia*, the monthly sample size was obtained using the formula $n = 4/prev$, where $n$ was the fish sample size, *4* originated from –Ln ($\alpha^*$sensitivity of the diagnostic method) and *prev* was the prevalence of the fish population [43]. The sample size values for *C. bonito* and *Ch. quaternia* were 5 and 13 animals, respectively. Thus, the monthly sample size of the two species for this study was 13 specimens.

## Parasitological examination

Due to the dynamics of commercial activity in the fishing port, fresh *C. hippurus* landed by the artisanal fishers were revised *in situ* while they were eviscerated and sold. For this reason, only the total length (TL) of each fish was recorded, and copepods were collected from both the opercular and buccal cavities, whereas the gills were stored in plastic bags, labelled and transported in ice-coolers to the laboratory for further microscopic examination. The parasites were washed in physiological saline solution (0.9%) and fixed in 96% ethanol. For taxonomic identification, copepods were cleared in increasing concentrations of glycerol/70% ethanol (1:20; 1:10; 1:5; 1:2), mounted on slides, and observed under optical microscopy to identify them using taxonomic keys [44,45].

The total body length of the 290 ovigerous females (OF) and the length of the 20 eggs of *C. bonito*, collected in both periods, were measured in milimetres with an ocular micrometre under a 10x magnification. Additionally, we calculated the gender proportion in both periods.

## Environmental variables

To describe the effects of EN on the prevalence and intensity of *C. bonito* and *Ch. quaternia* in dolphinfish, we included the Oceanic Niño Index (ONI) as an environmental variable; it is one of the most commonly used parameters to measure the EN and La Niña events. This index represents the monthly average values of the SST for the months before and after the normal conditions, which are then compared with the normal SST of the current month [46]. The EN

variability is obtained from the ONI from the 1+2 region, which is the smallest and eastern most EN regions (0–10˚S, 90–80˚W) and corresponds to the region of coastal South America. To indicate the EN conditions, the ONI of 3.4 region must be +0.5 ˚C or higher for at least five consecutive months, indicating that the east-central tropical Pacific is significantly warmer than usual. Values of ONI 1+2 were obtained from the National Weather Service Climate Prediction Center NOAA.

Values of SST, salinity and Chlorophyll *a* (Chl-*a*) were obtained through the R statistical package [47], using the ´xtracto 3D´ script [48]. The ´xtracto3D´ tool allows direct access to SST, salinity and Chl-*a* sensors [49]. The sensors used were the Advanced Very High Resolution Radiometer (AVHRR) for SST, the Hybrid Coordinate Ocean Model (HYCOM) for salinity and the Moderate Resolution Imaging Spectroradiometer (MODIS) for Chl-*a*. Monthly SST, salinity and Chl-*a* data were used to calculate values of these variables from the inshore fishing area (2˚N;79˚W –4˚S;79˚W and 2˚N;83˚W–4˚S;83˚W). Environmental variables were used to explain the variability of prevalence and the intensity of parasitism in the dolphinfish during the EN of 2015 to 2016.

## Data analysis

Dolphinfish sizes were grouped into 10 cm TL bins to determine whether the size distribution of the hosts varied between non-EN and EN periods. Differences in host size between non-EN/EN periods and seasons (dry and rainy) were tested with two-way analysis of variance (ANOVA). The prevalence (percent of infected hosts for each parasite species) and the mean intensity (the mean number of a particular parasite species per infected host) were calculated monthly [50].

To determine the relationship between the environmental variables (SST, salinity and Chl-*a*) and the infection parameters (prevalence and mean intensity) of *C. bonito* and *Ch. quaternia* in dolphinfish, we used the generalised additive models for location of scale and shape (GAMLSS) [51]. Additionally, multicollinearity of variables was evaluated through a variance inflation factor index (VIF) for which the usdm package of R was used. To decide which variables were discarded, a threshold of VIF < 4 was established and considered in the GAMLSS analysis [52]. The models were obtained assuming a normal distribution for prevalence and log normal distribution for mean intensity. The Akaike information criterion (AIC) value in the model setting GAMLSS package in R was used to fit the models. The best statistical model was selected by performing a forward procedure using the stepGAIC (generalised Akaike information criterion) function in the GAMLSS package, as this assessed the contribution of each variable and their combinations in the final model through an iterative process. This function chooses the best model based on the lowest AIC value. In addition, the power of the fit of each model was evaluated through the explained deviance (ED), expressed as a percentage [51].

Spectral analyses by Fourier series [53] were used to extract temporal variability patterns and periodical cycles of TL, SST, salinity, Chl-*a*, and infection parameters of *C. bonito* and *Ch. quaternia* in *C. hippurus*, along with the monthly patterns of ONI 1+2 (Statistica v.6 Statsoft©). This analysis required data points equally spaced in time [12,13,54]. Each temporal data set was transformed into sine curves of the same amplitude or harmonic frequencies [55] and it was represented in a periodogram. The harmonic frequencies, measured as spectral densities (strength of the frequency signal), represented a temporal scale of maximum variability in the temporal distribution [56]. Any marked frequency peaks were interpreted as a temporal scale of maximum variability to show trends of infection parameters and environmental variables. Spectral density values of parasite infection parameters and environmental variables were

compared using cross-correlation coefficients. The cross-correlation quantifies the temporal associations between variables and provides a measure of the similarity between two different data sets, determining the extent to which data sets exhibit correlated periodic variations. Time lags refer to the delayed responses of dependent variables; lag 0 corresponds to immediate response [57]. The time lag with the highest correlation coefficient is taken as the accurate time lag between the two-time series [58]. The cross-correlation coefficients were calculated for a significance of $p < 0.05$ [59]. The differences in the total length of OF, egg length and sexual proportion between non-EN and EN were evaluated with the Kruskal–Wallis test. These analyses were performed in Statistica v.6 Statsoft$^{©}$. Environmental and biological values were shown as minimal and maximal values (mean ± standard deviation).

## Ethics statement

The permit necessary to carry out parasite sampling and collection was obtained from the Ministerio de Ambiente del Ecuador (permit number 011 JMC-DPAM-MAE). The target species is not endangered or protected, and the samples were obtained from the artisanal fishery of dolphinfish. There were no additional ethical considerations linked to this research.

## Results

We collected 956 specimens of dolphinfish *C. hippurus*, 674 in the non-EN period, with TL from 51.4−135 cm (mean 72.88 ± 12.45 SD), and 282 in the EN period, with TL from 50.2−136 cm (74.54 ± 15.64) (Fig 1A). Host size did not significantly differ between non-EN and EN periods ($F_{1, 953} = 2.99$; $p > 0.05$). In non-EN, the mean host size in rainy season (79.47 ± 0.92 cm) was significantly higher than dry season (69.48 ± 0.64 cm); similarly, in EN, the mean host size in rainy season (81.40 ± 1.32) was significantly higher than that in dry season (72.04 ± 0.76) ($F_{3, 951} = 40.06$; $p < 0.05$). A post hoc test showed that host size was not significantly different between the rainy seasons for the non-EN and EN periods, but differed between the dry seasons for the non-EN and EN periods. The size distribution of *C. hippurus* presented a unimodal distribution with modes of 60–69.9 cm, following the same trend for the non-EN and EN periods (Fig 1B–1C); however, there was a high number of larger dolphinfish (> 80 cm) during the EN period (Fig 1C). The TL fluctuated temporally, with a peak of maximum variability every 11 months (Fig 1D).

Infection parameters of *C. bonito* and *Ch. quaternia* were significantly correlated with ONI 1+2, SST, TL and Chl-*a* throughout the GAMLSS model, and ED contribution ranged from 16–36% (Table 1). Salinity was not a predicting variable for infection parameters of both copepod species.

The SST ranged 21.30–26.95˚C (24.16 ± 1.68˚C) (Fig 2A), salinity 32.7–34.74 Practical Salinity Units (PSU) (33.58 ± 0.60 PSU; Fig 2B), Chl-*a* from 0.49–1.37 µg L$^{-1}$ (0.85 ± 0.25 µg L$^{-1}$; Fig 2C) and ONI 1+2 from -0.78–2.87 (1.0 ± 1.19; Fig 2A–2C and S1 Appendix). SST was not significantly different between seasons ($F_{1, 20} = 0.58$; $p > 0.05$) in both periods, but it was significantly higher during EN (25.3˚C) than non-EN (23.5˚C) ($F_{1, 20} = 13.24$; $p < 0.05$). Salinity was not significantly different between seasons ($F_{1, 22} = 0.24$; $p > 0.05$) in both periods, but it was significantly higher during EN (34.1 PSU) than non-EN (33.3 PSU) ($F_{1, 22} = 15.50$; $p < 0.05$). Chl-*a* was significantly higher in rainy (1.01 µg L$^{-1}$) than dry season (0.71 µg L$^{-1}$) ($F_{1, 13} = 9.72$; $p < 0.05$) in non-EN period, but not between seasons ($F_{1, 7} = 4.97$; $p > 0.05$) during EN period.

The SST, Chl-*a*, salinity and ONI 1+2 fluctuated temporally (Fig 2A–2C), with a peak of maximum variability every 11 months (Fig 2D–2E). Additionally, we found a significant positive cross-correlation between SST, salinity and Chl-*a* with ONI 1+2 with no lag, thus showing an immediate response of environmental variables resulting from the change in ONI 1+2.

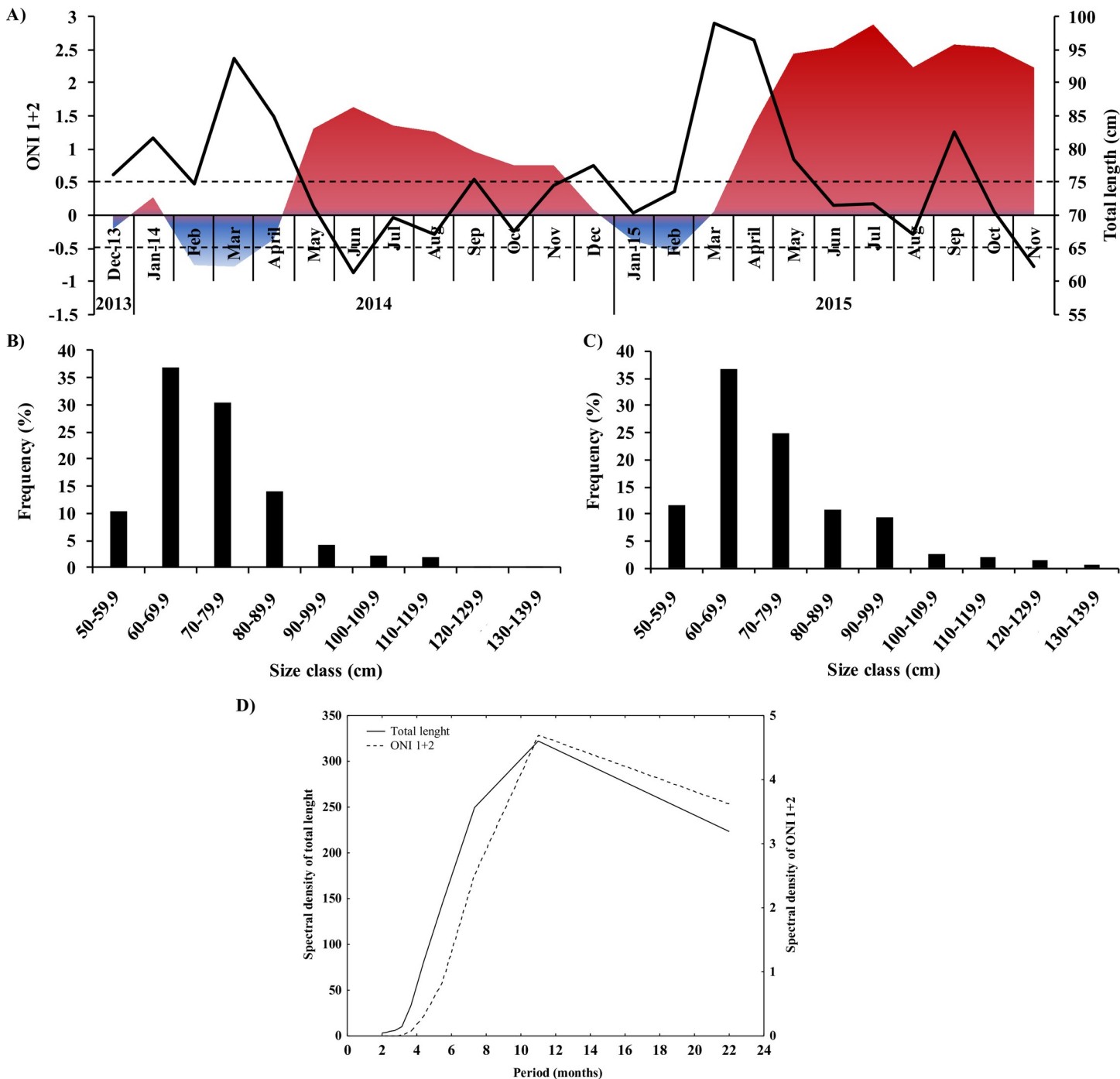

**Fig 1. The temporal fluctuation of total length for *Coryphaena hippurus* and size-frequency during non-El Niño (non-EN) and El Niño (EN) periods.** (A) total length (TL) and Oceanic Niño Index (ONI 1+2), size structure (size classes of 10 mm intervals) in (B) non-EN periods and (C) EN periods, (D) spectral density of TL (black line) and ONI 1+2 (dotted line).

The time series of the infection parameters of *C. bonito* showed considerable variation over the 2-year period; prevalence fluctuated from 29%-100%, with a mean intensity of 2.4–14.36 (6.73 ± 2.82) individual parasites per fish (Fig 3A and 3B and S2 Appendix). Spectral analysis of prevalence and mean intensity of *C. bonito* showed peaks of high variability every 11 months

**Table 1. The general additive models (GAMLSS) for the prevalence and mean intensity of *Caligus bonito* and *Charopinopsis quaternia* in *Coryphaena hippurus*.**

| Model | Df | Global deviance | Percent of explained deviance | Akaike criterion | *p* |
|---|---|---|---|---|---|
| Prev Cb ~ cs(ONI 1+2) + cs(SST) + cs(Chl-*a*) + cs(TL) | 18 | 162.72 | 21.9 | 198.72 | *p* < 0.001 |
| Int Cb ~ cs(ONI 1+2) + cs(Chl-*a*) + cs(TL) | 14 | 71.65 | 36.18 | 99.65 | *p* < 0.001 |
| Prev Cq ~ cs(ONI 1+2) + cs(Chl-*a*) + cs(SST) + + cs(TL) | 18 | 162.72 | 16.84 | 198.72 | *p* < 0.001 |
| Int Cq ~ cs(ONI 1 +2) + cs (SST) + cs (Chl-a) + cs(Lt) | 18 | 101.15 | 16.28 | 137.16 | *p* < 0.001 |

Abbreviations: Prev Cb: prevalence of *Caligus bonito*; Int Cb: mean intensity of *Caligus bonito*; Prev Cq: prevalence of *Charopinopsis quaternia*; Int Cq: mean intensity of *Charopinopsis quaternia*; ONI 1+2: Oceanic El Niño Index region 1+2; SST: sea surface temperature; Chl-*a*: chlorophyll-*a*; TL, total length of *Coryphaena hippurus*; Df: degrees of freedom; cs: cibic splines.

(Fig 3C), as observed for SST, Chl-*a*, salinity and ONI 1+2 (Fig 2D–2E). Positive cross-correlation among prevalence, mean intensity, Chl-*a* and ONI 1+2 were observed at lag 0. A significant association was revealed between prevalence and mean intensity with SST with no lag (Fig 3D).

The infection parameters of *Ch. quaternia* fluctuated over the 2-year period, with prevalence ranging 14%-65% and mean intensity from 2.33−9.28 (5.5 ± 1.86) (Fig 4A–4B and Table 2). Spectral analysis of these two parameters showed a peak of high variability every 3 and 11 months, respectively (Fig 4C). Cross-correlation analysis showed a significant association of prevalence and mean intensity of *Ch. quaternia* with ONI 1 + 2, Chl-*a* and SST with a 4-month lag (Fig 4D). Significant cross-correlation was found between the mean intensity of *Ch. quaternia* and SST with no lag (Fig 4D).

The results of the Kruskal–Wallis tests showed that egg length (KW-H$_{(1, 1214)}$ = 88.16; $p < 0.05$) and gender proportion (KW-H$_{(1, 83)}$ = 3.76; $p < 0.05$) were significantly different between non-EN and EN periods. In the latter, eggs were larger, and females more abundant (Fig 5A). The total body length of OF showed no significant differences between non-EN and EN periods (KW-H$_{(1, 292)}$ = 1.05; $p > 0.05$) (Fig 5B); however, there was a clear tendency to larger females in EN periods.

## Discussion

Changes in the infection parameters of copepods are explained by the variation of those oceanographic (SST, Chl-*a* and ONI 1+2) and biological variables (TL), associated to EN event. This climatic phenomenon is characterised by warmer values of the equatorial Pacific SST altering both physical and biological oceanography. Evidence has demonstrated that ENSO can modify the infection parameters of parasites in marine organisms; however, these studies have only been carried out in the temperate latitudes [6,17,19,20]. Although our results come from a short time series, they allow detecting periodic changes in oceanographic variables and infection parameters. This is the first tropical survey to provide evidence that increased SST promotes higher infection parameters of two species of parasitic copepods.

### The host factor

Host length showed an annual pattern, with larger individuals occurring in the rainy seasons and smaller ones in the dry seasons, which was consistent during the EN and non-EN periods. These peaks might be associated with migration of large and small fish individuals during reproductive periods and/or favourable conditions for prey availability in the area. Seasonal peaks of fish abundance and size, associated with variable SST, have been widely documented in dolphinfish from the Atlantic and Pacific Oceans [60,61,62,63,64,65,66,67,68], which

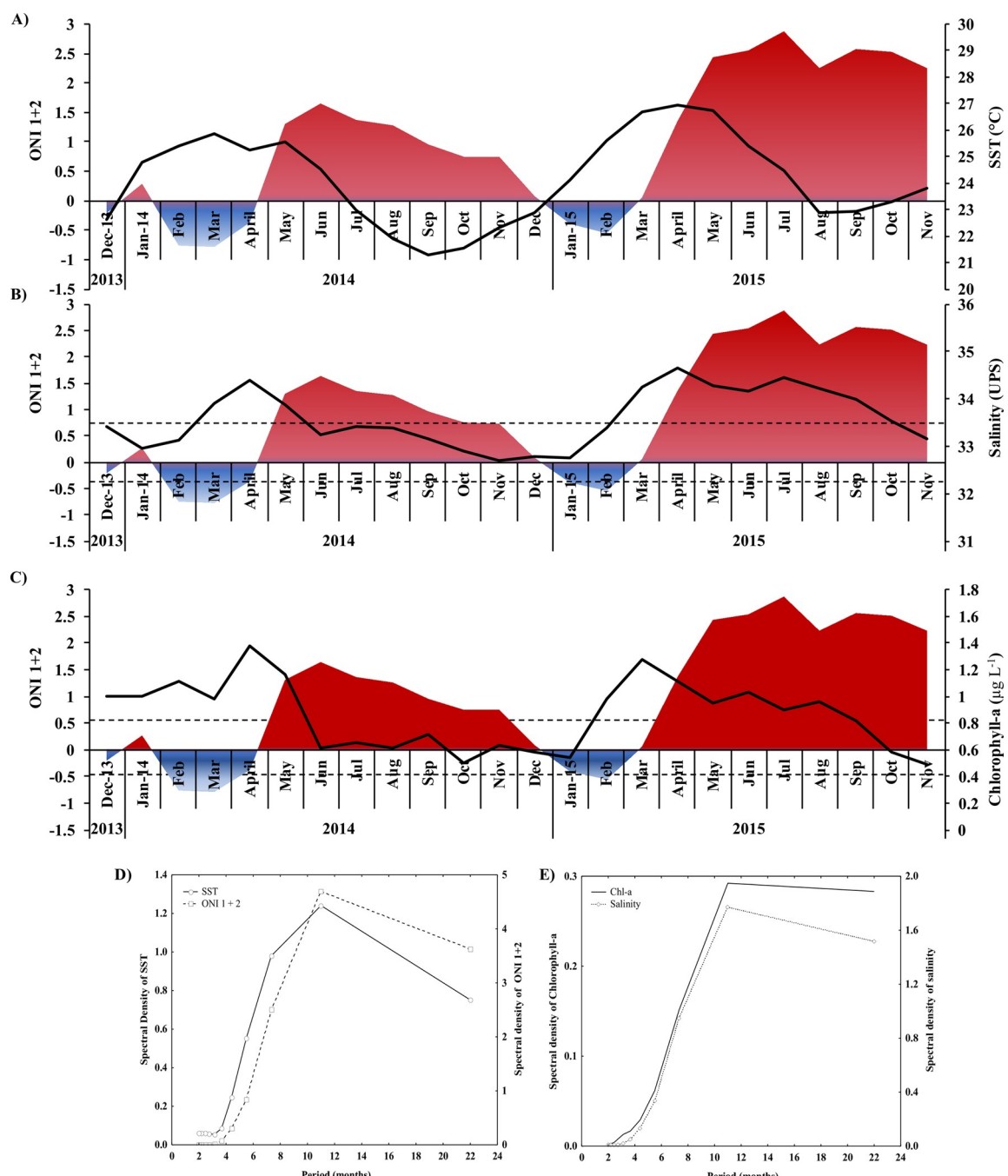

**Fig 2. The temporal fluctuation of environmental variables of the Tropical Eastern Pacific from December 2013 to November 2015.** (A) sea surface temperature (SST) and Oceanic Niño Index (ONI 1+2) fluctuation, (B) salinity and ONI 1+2 fluctuation, (C) chlorophyll-*a* (Chl-*a*) and ONI 1+2 fluctuation, (D) the spectral density of SST (black line) and ONI 1+2 (dotted line) by Fourier series (*x*-axis in months), (E) the spectral density of Chl *a* (black line) and salinity (dotted line) by Fourier series (*x*-axes in months).

indicates the presence of at least two cohorts across the year associated with pre-spawning migration or favourable conditions, such as thermal fronts [60,62,69]. It has mentioned that the dolphinfish exhibits bimodal reproductive behaviour, thus they apparently have different size groups arriving at different time periods across the year [67,70]. Size distribution was

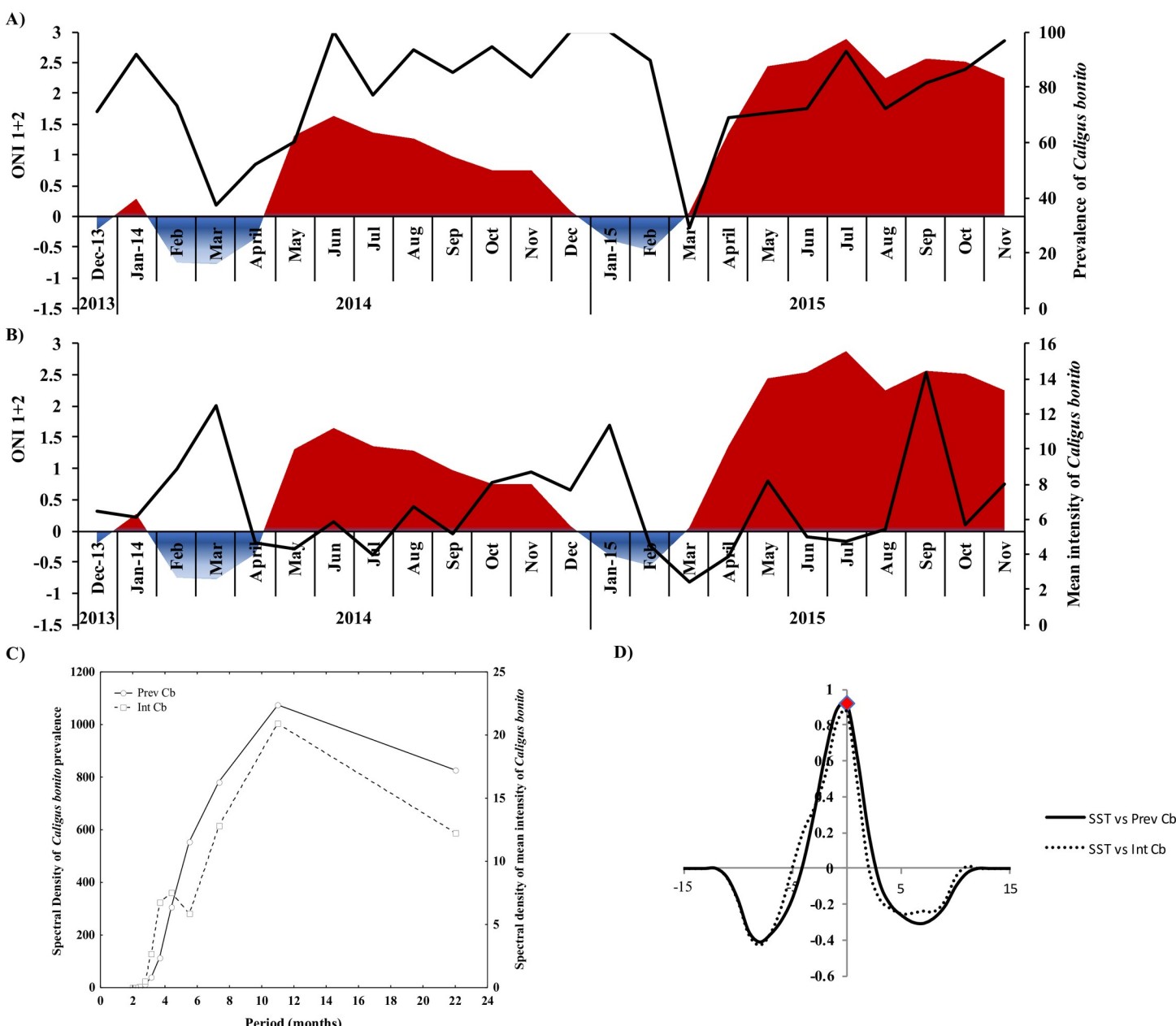

**Fig 3. The temporal fluctuation (2013–2015) of prevalence and mean intensity of *Caligus bonito* in *Coryphaena hippurus* from the Tropical Eastern Pacific.** (A) prevalence and Oceanic Niño Index 1+2 (ONI 1+2), (B) the mean intensity and ONI 1+2, (C) the spectral density of prevalence (black line) and mean intensity (dotted line) by Fourier series, (D) cross-correlations between the prevalence and mean intensity of *C. bonito* and ONI 1+2.

similar between non-EN and EN periods, although a higher number of larger specimens (> 75 cm) appeared during the latter, thus causing a peak in September. The recruitment of larger animals did not change the seasonal pattern during the year in both periods (non-EN and EN), but promoted significant differences between the dry seasons of such periods. The occurrence of larger individuals of *C. hippurus* could be related to the increase in SST during the EN period, because of the preference of species for the warm water pool. Two of the most important oceanographic variables correlated with the high catching rates in the Atlantic and Pacific Oceans are warm SST and low levels of surface chlorophyll-*a* concentration [71,72,73]. These

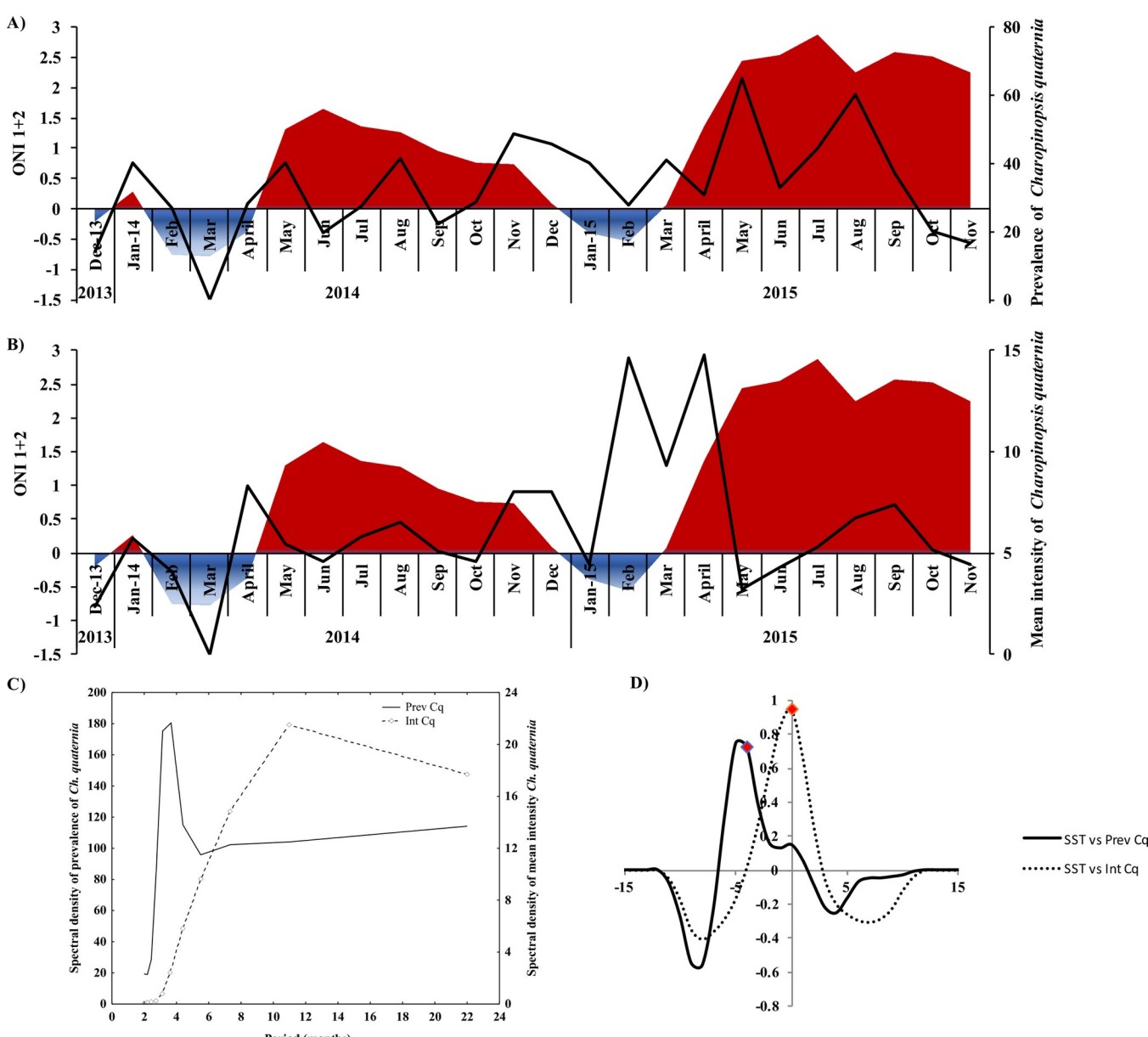

**Fig 4. Temporal fluctuation (2013–2015) of prevalence and mean intensity of *Charopinopsis quaternia* in *Coryphaena hippurus* from the Tropical Eastern Pacific.** (A) prevalence and Oceanic Niño Index 1+2 (ONI 1+2), (B) mean intensity and ONI 1+2, (C) the spectral density of prevalence (black line) and mean intensity (dotted line) by Fourier series, (D) cross-correlations between the prevalence and mean intensity of *Ch. quaternia* with ONI 1+2.

conditions coincided with EN periods that favoured the greater landings of dolphinfish during EN events of 1983, 1987 and 1998 [74].

## The environmental factors

We quantified an annual pattern of the oceanographic variables (Chl-*a*, salinity and SST) at roughly 11-month cycles, where the two known climatic seasons; rainy (wet/warm) from December to May and dry (dry/cold) season from June to November were recognized. This time series (24 months) was related to the complex, highly dynamic oceanography of TEP,

**Table 2. Infection parameters of copepod *Caligus bonito* and *Charopinopsis quaternia* parasitizing *Coryphaena hippurus* from the Tropical Eastern Pacific.** In parenthesis number of fish revised.

| | *Caligus bonito* | | *Charopinopsis quaternia* | |
|---|---|---|---|---|
| | Prevalence | Mean intensity | Prevalence | Mean intensity |
| December 2013 (21) | 71 | 6.46 ± 5.08 | 14 | 2.33 ± 2.31 |
| January 2014 (62) | 92 | 6.1 ± 4.14 | 40 | 5.76 ± 7.73 |
| February 2014 (71) | 73 | 8.85 ± 4.22 | 27 | 4.1 ± 6.61 |
| March 2014 (9) | 38 | 12.5 ± 0.7 | - | - |
| April 2014 (25) | 52 | 4.69 ± 5.92 | 28 | 8.28 ± 7.78 |
| May 2014 (60) | 60 | 4.3 ± 6.86 | 40 | 5.41 ± 2.87 |
| June 2014 (51) | 100 | 5.86 ± 3.89 | 20 | 4.6 ± 6.15 |
| July 2014 (22) | 77 | 4 ± 3.2 | 27 | 5.83 ± 10.4 |
| August 2014 (75) | 93 | 6.68 ± 4.89 | 41 | 6.55 ± 11.66 |
| September 2014 (68) | 85 | 5.16 ± 5.03 | 22 | 5.06 ± 5.99 |
| October 2014 (74) | 95 | 8.08 ± 5.41 | 28 | 4.57 ± 7.64 |
| November 2014 (43) | 84 | 8.72 ± 4.93 | 49 | 8 ± 12.81 |
| December 2014 (33) | 100 | 7.69 ± 5.02 | 45 | 8 ± 8.80 |
| January 2015 (30) | 100 | 11.33 ± 8.34 | 40 | 4.41 ± 5.66 |
| February 2015 (29) | 90 | 4.46 ± 2.90 | 28 | 14.62 ± 20.80 |
| March 2015 (17) | 29 | 2.4 ± 2.19 | 41 | 9.28 ± 12.06 |
| April 2015 (13) | 69 | 3.88 ± 3.05 | 31 | 14.75 ± 26.83 |
| May 2015 (17) | 71 | 8.16 ± 6.32 | 65 | 3.18 ± 3.4 |
| June 2015 (58) | 72 | 5 ± 3.77 | 33 | 4.26 ± 2.64 |
| July 2015 (43) | 93 | 4.75 ± 3.12 | 44 | 5.31 ± 7.20 |
| August 2015 (40) | 73 | 5.41 ± 4.06 | 60 | 6.71± 7.58 |
| September 2015 (27) | 81 | 14.36 ± 9.88 | 37 | 7.4 ± 10.34 |
| October 2015 (30) | 87 | 5.69 ± 3.71 | 20 | 5.16 ± 9.72 |
| November 2015 (30) | 97 | 7.97 ± 5.29 | 17 | 4.4 ± 5.64 |

influenced by the Humboldt Current and Equatorial countercurrent. It is recognized that the oceanic environment in the TEP varies seasonally, inter-annually and on larger time scales (decadal and multi-decadal) [75]; therefore, an extended time series would be required to obtain stronger data comprising a more extensive range of interannual variations. The Humboldt Current is often described as carrying cold, nutrient-rich waters northward, promoting upwelling processes off Ecuador, Peru and northern Chile. This current recedes southwards around December each year resulting from the flow of warm waters of the South Equatorial countercurrent [40,76]. The SST, salinity and Chl-*a* values were higher during rainy than dry season in both years, but the first variable increased during the dry season of 2015 and showed a positive correlation with ONI 1+2. This pattern was previously observed in TEP, where EN modified the Humboldt Current System, affecting different environmental variables (i.e., Chl-*a*, salinity and SST), as well as the distribution of Pacific fish and squid [77,78,79,80,81].

## Infection parameters annual patterns

The infection parameters of *C. bonito* and *Ch. quaternia* showed an annual behaviour on a roughly 11-month cycle. This annual pattern is characterised by lower average values the rainy season than in dry season, which is inverse to the TL pattern of host. Although host size is frequently recognised as a determinant of parasite abundance (as larger hosts may harbour higher numbers of individual parasites than smaller hosts) [82], our findings showed an opposite

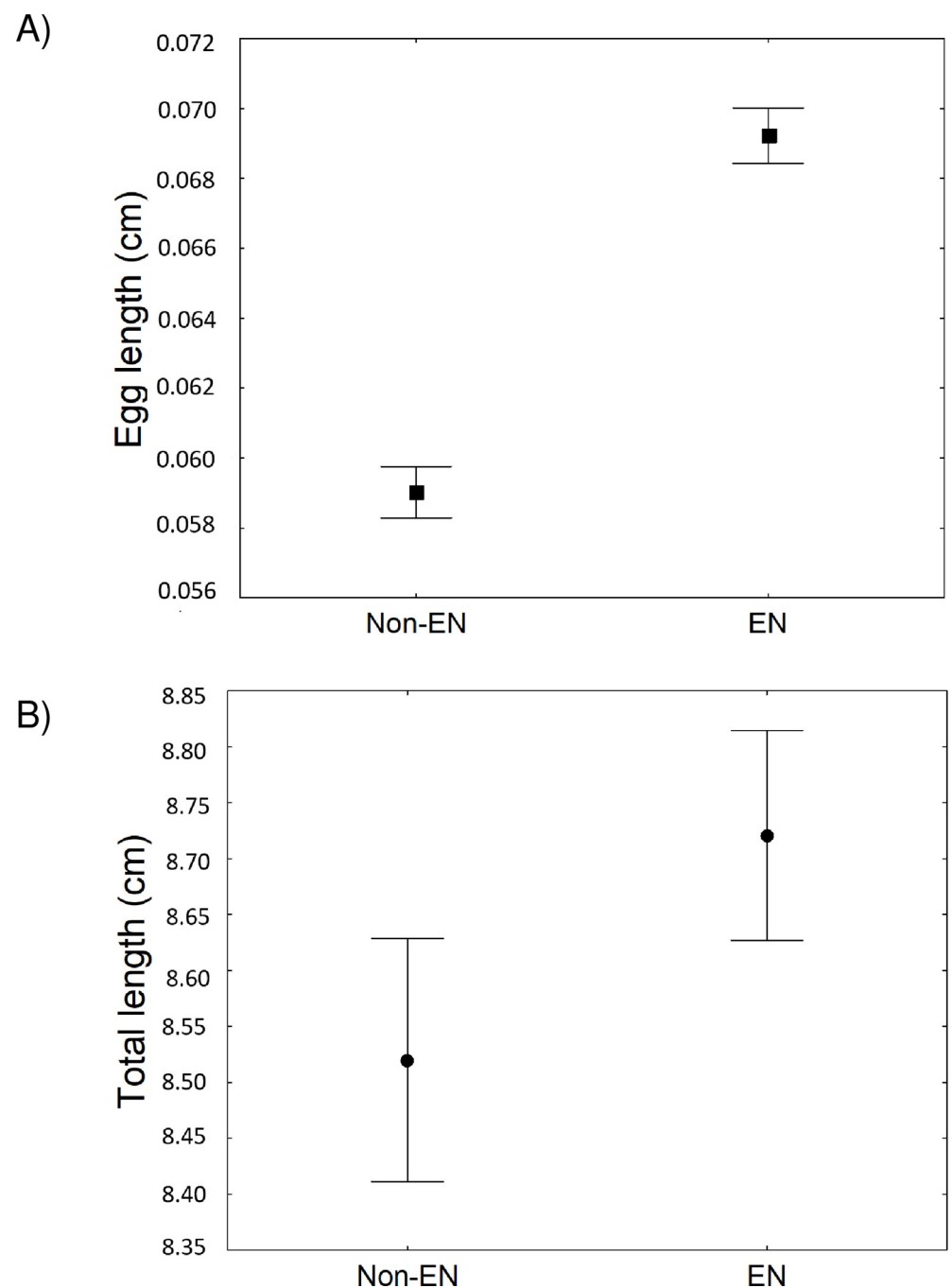

**Fig 5. Egg length and the total length of ovigerous females of *Caligus bonito* in relation to non-El Niño (non-EN) and El Niño (EN) periods.** (A) egg length, (B) total length of ovigerous females.

pattern, as seen in other hosts [83,84,85]. This may be a consequence of at least three factors: (1) host immunity; larger fish have more efficient immune resistance than younger fish as a defence mechanism against external infections [86], thus reducing the chances of parasite infections; (2) two cohorts with variations in their relative abundances of parasitic taxa. Apparently, two cohorts of dolphinfish are present every year in both the Eastern Central Pacific and the Atlantic Ocean [69,87,88], which could reflect variations in infection parameters between

size classes or distribution areas, due to their characteristic of segregating by size and sex [88,89,90,91]; and (3) the migration pattern of host can affect parasite dynamics. The dolphin-fish is a highly vagile species with migrations conditioned by the SST (20°C) and by its repro-duction pattern, voracious appetite and availability of floating objects [38,89]. Thus, habitats with different environmental variables and host diversity and abundance might affect the prevalence and abundance of parasites. These hypotheses could be clarified when the biology of *C. hippurus* in the Eastern Central Pacific is studied.

The infection parameters of *C. bonito* and *Ch. quaternia* were explained by TL, Chl-*a*, SST and ONI 1+2, indicating that EN had an effect on the infection parameters of both species. The increased SST modified the infection parameters of both copepod species and other envi-ronmental variables, mainly in the dry season during EN, thus altering the seasonal cycle (dry-rainy). However, higher prevalence and mean intensity in *Ch. quaternia* and mean intensity in *C. bonito* were more evident during EN; whereas prevalence of the latter copepod species raised from May 2014 (non-EN period) when ONI 1+2 was <0.5°C higher than usual. Enhanced prevalence of *C. bonito* might be due to its preference for warm water that increases its abundance in the tropics [91,92]. Even though from June to November 2014 was a non-EN period, SST was higher than the historical average values (1982–2016) for those months (21.3 –24.5°C vs. 20.5–23°C, respectively) [93], thus indicating that the East-central Tropical Pacific was significantly warmer than usual (S2 Appendix). Alterations on abiotic factors can directly influence the infection dynamics of parasites [94], mainly in ectoparasites with direct life cycles (e.g., crustaceans, copepods and monogeneans) and in close contact with the environment [15]. High temperatures are often associated with increased frequency or severity of infection, as a result of altered pathogen development and survival, physiological changes and range expansion of the host [95]. For example, the generation time of sea lice (*Caligus rogercresseyi*) ranges from 50 days at 12°C to 114 days at 7°C; a shorter generation time has been related to warmer temperatures [96].

The association between the prevalence of *Ch. quaternia* with SST, TL and Chl-*a* showed a 4-month lag; in contrast, mean intensity showed an immediate response. An increase in inten-sity occurred first (January 2015), and several months later (May 2015), when prevalence raised, suggesting that infected fish showed higher infection rates until the parasites began to infect new hosts. Since *Ch. quaternia* and *C. bonito* shared the site of infection [97], it is likely that EN triggered an increase in their numbers per infected host, which could compete for space and spread to other hosts after some time. Although copepod abundance depends on their intrinsic fecundity and rates of growth and development, density-dependence intra- and inter-specific competition, as well as host responses to infestation, should also be considered [98].

## Copepods as parasites in changing conditions

Higher infection levels could be related to increases in egg length, female numbers and the total length of the OF in EN periods. Most studies on the effect of temperature on the repro-ductive parameters of copepods have been carried out in temperate species, and very few for tropical groups, like *Caligus* [98,99]. It is assumed that larger copepod females produce larger eggs, thus increasing survival of nauplii and improving their chances of reaching suitable hosts [100,101]. Another important factor is the increased rate of infecting larval stages of parasitic copepods at higher temperatures [102,103], which causes higher infection rates [101,104,105]. For example, this effect has been documented in the parasitic branchiuran *Argulus coregoni* Thorell, 1865 infecting *Salmo trutta* Linnaeus, 1758 populations from Finland [105]. Thus, with generalist parasites such as *C. bonito* [35], with a greater availability of oceanic hosts and both strong and long-term shifts towards favourable environmental conditions, we expect an

increase in the infection parameters of *C. bonito*. These results contribute to a deeper knowledge of copepods, mainly caligids, in tropical waters. However, further studies assessing the effects of global climate change or oceanographic phenomena (e.g. EN) on the reproductive dynamics of parasitic copepods and their effects on hosts are required.

## Parasites and global warming

Future ENSO events will be stronger and more frequent due to currently recognised global warming trends [1]. In recent years, NOAA has registered record SSTs at a global level and predicts that SST will increase throughout the rest of this century [106]. The models proposed by the Intergovernmental Panel on Climate Change (IPCC) predict that SST will be 1.1–2.9 ˚C higher by 2100, based on the most conservative B1 emission scenario, and by 2.0–5.4 ˚C higher, based on the more realistic A2 scenario [107,108]. The present study showed the positive relationship between EN in the Ecuadorian Pacific and infection parameters of two species of parasitic copepods. In the context of global climate change, the increase in water temperatures will influence the emergence and re-emergence of diseases, with adverse effects on hosts, modifying their spatial range and seasonal abundance, and causing a rupture in biological interactions. These changes could have significant ecological and economic consequences, especially in relevant fishing resources, such as *C. hippurus* and other fish species of the Eastern Pacific. Therefore, it is necessary to carry out long-term monitoring of this kind of parasite-host model to extrapolate the effects of environmental shifts on those interactions. We also recommend evaluating the effect of environmental variables on the infection parameters of tropical parasitic copepods under experimental laboratory conditions.

## Supporting information

**S1 Appendix. Oceanographic variables from ONI 1+2 region during December 2013 to November 2015.**
(DOCX)

**S2 Appendix. Average SST in the Niño (1+ 2) region, base periods of 30 years.**
(DOCX)

## Acknowledgments

The authors thank Víctor Caña-Bozada from Universidad Laica Eloy Alfaro de Manabí, and Andrea Mogro Mendoza and Carlos Mendoza Zambrano from Universidad Técnica de Manabí for field and laboratory support. We are grateful to the anonymous referees for reviewing and providing helpful comments.

## Author Contributions

**Conceptualization:** Ana María Santana-Piñeros, Yanis Cruz-Quintana.

**Data curation:** Ana María Santana-Piñeros, Yanis Cruz-Quintana, Geormery Mera-Loor.

**Formal analysis:** Ana María Santana-Piñeros, Ana Luisa May-Tec.

**Funding acquisition:** Ana María Santana-Piñeros, Eduardo Suárez-Morales.

**Investigation:** Ana María Santana-Piñeros, Yanis Cruz-Quintana, Geormery Mera-Loor.

**Methodology:** Ana María Santana-Piñeros, Yanis Cruz-Quintana, Ana Luisa May-Tec, Geormery Mera-Loor.

**Project administration:** Ana María Santana-Piñeros.

**Resources:** Ana María Santana-Piñeros, Yanis Cruz-Quintana.

**Supervision:** Eduardo Suárez-Morales, David González-Solís.

**Validation:** Eduardo Suárez-Morales, David González-Solís.

**Visualization:** Ana María Santana-Piñeros.

**Writing – original draft:** Ana María Santana-Piñeros, Ana Luisa May-Tec.

**Writing – review & editing:** Yanis Cruz-Quintana, María Leopoldina Aguirre-Macedo, Eduardo Suárez-Morales, David González-Solís.

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
