## [Decision Letter · Decision Letter 0]

10 Aug 2019

PONE-D-19-16590

The 2015-2016 El Niño increased infection parameters of copepods on Eastern Tropical Pacific dolphinfish populations

PLOS ONE

Dear Dr. Santana-Piñeros,

Thank you for submitting your manuscript to PLOS ONE. After careful consideration, we feel that it does not fully meet PLOS ONE’s publication criteria as it currently stands. Therefore, we invite you to submit a revised version of the manuscript that addresses the points raised during the review process.

As you will see, there are diverse opinions among reviewers regarding your study. However, there are two major concerns shared by all reviewers and myself: your study present correlative evidence of an association between parasite infection rates and El Niño temperatures, which cannot be used as an evidence of causality. You would need proper controls, longer time series, comparisons across species or experimental work to support your conclusion, otherwise you need to clearly state the limitations of your work. Also, your manuscript must be revised by an English native speaker before submitting your revised version.

We would appreciate receiving your revised manuscript by Sep 24 2019 11:59PM. To enhance the reproducibility of your results, we recommend that if applicable you deposit your laboratory protocols in protocols.io, where a protocol can be assigned its own identifier (DOI) such that it can be cited independently in the future. For instructions see: http://journals.plos.org/plosone/s/submission-guidelines#loc-laboratory-protocols

We look forward to receiving your revised manuscript.

Kind regards,

Jose M. Riascos, Ph.D.

Academic Editor

PLOS ONE

Journal Requirements:

Reviewers' comments:

Reviewer's Responses to Questions

**Comments to the Author**

1. Is the manuscript technically sound, and do the data support the conclusions?

Reviewer #1: Yes

Reviewer #2: Partly

Reviewer #3: Partly

2. Has the statistical analysis been performed appropriately and rigorously? 

Reviewer #1: Yes

Reviewer #2: I Don't Know

Reviewer #3: I Don't Know

3. Have the authors made all data underlying the findings in their manuscript fully available?

Reviewer #1: Yes

Reviewer #2: Yes

Reviewer #3: Yes

4. Is the manuscript presented in an intelligible fashion and written in standard English?

Reviewer #1: Yes

Reviewer #2: Yes

Reviewer #3: No

5. Review Comments to the Author

Reviewer #1: General comments:

Santana-Pineros et al present a relevant study showing heightened prevalence and infection load of ectoparasitic copepods on dolphinfish during the recent severe El Nino in the TEP. I think the manuscript could use a careful edit by a native English speaker, I tried to assist as much as I could with grammar and spelling. Also data availability statement is made in the submission form, but there is no text in methods or elsewhere specifying where and how one could obtain raw data. Also mentioned are supplemental files for data and maybe some methods? Results? yet no supplemental files were in the pdf for review, nor are they cited anywhere in the text. The abstract needs to include specific values from the results section, not just increase/decrease statements. Finally, some restraint is warranted in extrapolating the data observed to draw general conclusions. While there was a clear correlation between parasite infection rates and El Nino, without proper controls or comparisons across species or multiple El Nino events it is difficult to attribute causation. For instance, in Figure 3A, it would appear to me that C. bonito prevalence is highly seasonal, rather than being directly driven by the El Nino event.

Once these issues are dealt with, and specific line comments (below) are addressed, I think this will be a good contribution for publication in PloS One.

Specific comments:

Line 39: Since study is specific to TEP, maybe restrain intro to TEP

Line 41: Parasites respond to environmental variations, but he influence of ENSO cycles on the seasonal variation of parasitic copepods has not yet been evaluated.

Line 45: ... in the dolphinfish Coryphaena hippurus and oceanography during the 2015-16 El Nino event. *** ENSO is not synonymous with El Nino, which is just one state or extreme of the ENSO cycle (which also includes La Nina on the opposite side of the spectrum). When referring to this event, please use El Nino or EN, not ENSO.

Line 46: Fish were collected from capture fisheries on the Ecuadorian coast (Tropical Eastern Pacific - TEP)

Line 47: Variation in sea surface temperature

Line 52: Of these, only SST increased consistently between February and November 2015

Line 55: both copepod species and increasing SST.

Line 57: Please state actual values by which prevalence and intensity of C. bonito and Ch. Quaternia increased. Also is this truly the only study that has looked at parasitic copepod abundance relative to sea surface temperature variation? Statements such as this throughout the MS should be carefully double-checked or re-worded.

Line 64: El Nino-Southern Oscillation (ENSO) events create fluctuations in sea surface temperature (SST) >0.5 �C in the Tropical Eastern Pacific (TEP). During El Nino events,…

Line 68: waters have a profound influence on local weather, ocean conditions, and marine and terrestrial ecosystems.

Line 69: Consider moving sentence on line 75 that starts “it has been suggested…” to before this sentence, since it provides precedent for climate change connection (having not been mentioned before this point leaves the reader bewildered)

Line 77: ENSO affects host-pathogen relationships of some marine organisms by reducing host immunity; consider citing recent paper on ENSO-fish disease in Galapagos (Lamb et al 2018 – Scientific Reports)

Line 80: fungal, viral, protozoan, or internal metazoan parasites;… such as metazoan ectoparasites, which are more exposed and infliuenced…

Line 86: Parasitic copepod loads have long been recognized as a factor affecting the growth, fecundity, and survival of both wild and farmed fish… NOTE: here you used the oxford comma (comma before “and” at the end of a list), whereas at other points in the MS you do not. The choice is up to you but please strive for consistency.

Line 88: to feed on their mucus, tissues, and blood, and leave open wounds exposing the host to secondary infections, leading to increased mortality…

Line 98: populations could be better managed in the region…

Line 104: moving northwest and the South Equatorial Current…

Line 109: phytoplankton, zooplankton, and pelagic fish communities

Line 113: SOI is listed here without previous mention or acronym definition; needs explaining

Line 114: we used a 2-year time series of monthly sea surface temperature…

Line 116:mWith prevalence and intensity of parasites in dolphinfish…

Line 117: Tropical Eastern Pacific during the El Nino event of 2015-2016.

Line 121: ….47 W), in the city of Manta, Manabi, Ecuador

Line 148: To calculate the sample size, we calculated the prevalence of both C. bonito and Ch. Quaternia in 154 individual hosts during pre-El Nino years. (Is this correct? Please be more detailed and specific how/why these fish were used as a baseline. What were the oceanographic conditions during the pre-collection phase relative to EN year?

Line 153: minimum monthly sample size to achieve what? Detect a single infection? Again please be specific.

Line 160: in plastic bags, labelled, and transported… for further microscopic examination

Line 171: citation should be for raw data, not package used to summarize it. Please find out where extraco3D acquires its datasets and cite them directly.

Line 185: Dolphinfish sizes were grouped into 10 cm total length bins in order to determine whether the size distribution of hosts varied between non- El Nino and El Nino periods.

Line 193: with monthly values of SST, salinity…

Line 211: Did you measure length, weight? Can you look at length/weight ratios and whether they varied between non – EN and EN periods, or any other parameter of body condition or physical health that could otherwise show an overall health effect of EN and/or copepod infection?

Line 216: ENSO periods do not affect host sizes or the size distributions of dolphinfish populations

Line 224: SST increased steadily from February 2015 until the end of the study, while

Line 227: and SOI 1+2 with no lag

Line 229: environmental variabiles in the Tropical Eastern Pacific

Line 233: by Fourier series (x-axis in months); (C) Spectral density of sea surface temperature… Fourier series (x-axis in months)

Line 239: with a mean intensity of … individual parasites per fish.

Line 241: … as did SST, Chl-A

Line 243: were observed with no lag; NOTE: continue this usage throughout (instead of “at lag 0”)

Line 245: mean intensity and SOI 1+2 with no lag. …a shift of SST variability related to the influence of SOI 1+2 coincident with changes in Chl-a and salinity, all of which affected the infection parameters of… NOTE: since these oceanographic factors co-varied you cannot partition the effects of each in this MS

Line 253: of prevalence (black line) and infection intensity… of C. bonito and sea surface temperature… cross-correlations between prevalence and infection..

Line 259: fluctuated over the two year period, with prevalence ranging 14-65%, and mean intensity ranging …

Line 264: salinity, and SST at a 4-month lag (again, use this annotation throughout)

Line 267: salinity and SST with no lag (Fig 4C). Since SST, Chla-a, and salinity co-varied over the study period, they may have acted simultaneously to trigger increased infection levels of Ch. Quaternia on C. hippurus. Since these oceanographic variables are highly correlated, it was impossible to detect the effect of each factor alone. … intensity and SOI 1+2 with a 4-month lag.

Line 278: SST (C), chlorophyll a (Chl-a)… and the prevalence and infection intensity…

Line 281: (ENSO) is one of the most dominant and consequential climate cycles on Earth… The EN (or El Nino, but not ENSO) phase… altering both environmental and biological oceanography.

Line 287: A specific case where extrapolation goes to far; correlation between ENSO and copepod infection prevalence does not signify causation

Line 289: Although we quantified seasonal and annual patterns of several oceanographic variables, only SST… was associated with the El Nino event of 2015-2016… average temperature relative to a 136-year record… SST for the global ocean was… Ocean temperatures during the first three months of 2015 were each the third warmest on record for their respective months

Line 305: and Chl-a on a roughly 11-month cycle

Line 307: parasitic copepod species might respond synergistically to two events: the increase in the number of host individuals *NOTE: this is the first mention of increased population size during El Nino, it is not reported on or shown in figures or tables in the Results section. Or are you referring to the citation on line 317? If so that citation should be mentioned before this statement.

Line 312: This sentence is almost illegible and I wouldn’t try to rewrite since I am unsure of the context. For this reason and myriad other small edits needed, I would again recommend the MS be carefully proof-read by a native English speaker.

Line 320: We documented … copepod species and SST. Our observations suggest that.. an increase in the prevalence and mean intensity…

Line 324: SST probably caused a faster and sustained increase…

Line 327: Rising infection parameters could be related to the higher number of suitable hosts due to high temperature *NOTE: again tenuous since you did not show data to support this…rate of infection between hosts, or increased reproduction rates of parasites due to high temperature.

Line 332: parasite infection is host fish population… copepod parasites, proximity…

Line 338: thus, with generalist parasites, a greater availability of oceanic hosts and both short and long-term shifts towards favorable environmental conditions, we expect…

Line 343: four months, in contrast to mean intensity, which showed an immediate response… triggers an increase in the… per infected host, which… * NOTE: Any evidence for Ch. quaternia to be in competition for space on hosts? Please cite authority on this statement.

Line 349: Additionally, this lag (up to 4 months)

Line 357: NOAA has already registered record SSTs evaluated during the 1880-2016 period

Line 361: by the Intergovernmental…

Line 365: showed the positive relationship between ENSO… and infection parameters… increase in water temperatures will influence… with negative effects on host organisms, modifying spatial range and seasonal abundances, and causing a rupture in biological interactions.

Line 371: especially in relevant fishing resource species, such as C. hippurus and other species… necessary to carry out long-term monitoring of this kind of parasite-host model to extrapolate the effects of environmental shifts on parasite-fish interactions.

Line 384: sampling permit

Figure 1 Decimal values should be labelled with a period, not a comma

Reviewer #2: Copepods are not convincingly presented to be good bioindicators.

The sampling number in the el nino period is less than the non el nino period.

The discussion is presented speculatively and the presentation of information is not in a logical sequence.

The main conclusion of the manuscript is that the increase in temperature in the el nino causes an increase in the copepod population, but in the graph it is verified the increase in temperature gradually, in other words, the peaks in the el nino do not seem significant to affirm that they influence the copepod population.

Reviewer #3: Dear Authors,

I have been asked to review your manuscript ‘El Nino increased infection parameters of copepods on Eastern Tropical Pacific dolphinfish populations’ (PONE-D-19-16590). The idea of testing the response of parasites to climate variability is very interesting, and I appreciate the amount of work you have put in the study. However, I have some major concerns (and a few minor comments) that first need to be solved before this manuscript can be accepted for publication.

First of all, I am afraid that your conclusion is not entirely supported by your analyses. You conclude that tropical parasites are sensitive to climate variability. However, you support your conclusion entirely correlative ‘evidence’. When there is a correlation between two variables (like SST and infection parameters) it does not necessarily mean that the sea surface temperature is responsible for increases in infections. It would have been better to analyze this with general additive models so you have a better idea of causality. Furthermore, it would be good for the manuscript to see some additional prove. For example, an experiment that shows that infective larvae have higher development rates with higher temperatures. There has been some speculation about this in the discussion (for other copepod species), but a small experiment would give the reader more confidence in your conclusion. In the discussion there is also speculation about an increase in host abundance as a result of higher temperatures. Are there not data (fish landings for example) to support this idea? In fairness, the discussion that should support your conclusions is way to speculative for the reader to believe that your conclusions are true and what the prime cause of your observations could be.

Second, the English of the manuscript is not good enough for publication. I highly recommend giving your manuscript to a native English speaker before submitting it again. In the minor comments section I will give some examples of sentences that were unclear to me. Here and there I have found also some Spanish words.

Third, the figures are too crowded. Having four different y-axes instead of one is very confusing. Also, Figure 2D, 3E and 4D require more explanation.

MINOR COMMENTS

L69: Climate change or ENSO?

L75/76: Please use ‘the’ before ENSO

L77: Change host-pathogens relations to host-pathogen relationships

L78: Remove ‘the’ before host immunity

L78/79: Change the virulence of pathogens to pathogen virulence

L80: Reference 10 goes before 25 and 56 (remove Table 1)

L83-85: Delete. This is too specific and exactly what you are going to do in this study.

L86: Substitute “load has long been recognized as a factor affecting” by “are known to affect”

L89: Substitute “leave” by “thereby leaving”. Substitute “exposing the host to” by “that can result in”

L113: This is the first time you use the abbreviations, please write out.

L115/116: This is the second time you use abbreviatons, please do not write out

L116: What exactly is the Southern Oscillation index? Please explain

L120: A fishing port is not really a study or sampling site. It’s a site where you collect the fish. Please change the title to “Collection of fish hosts”

L126: Remove title

L127_134: How are the fish caught? With nets? Long lines? Is there a chance that the parasites are removed by the fishing method?

L146: How many fish per month?

L147/148: Why are the lengths of the ENSO and non-ENSO period different?

L148: Substitute “calculated” by “calculate”

L148-155: This is very unclear to me. I had to read reference 14 to understand what you did. Please be more detailed here.

L161: Remove “All gills were dissected under a microscope”. You said the same before

L164: Increasing concentrations of glycerol. Please be more specific.

L165: I assume that the slides were also examined under a microscope. Please add this information

L175/176: I really do not know what you mean with these values and regions, and why they are important for your study.

L178/178/182: The websites are not working, please update the links

L185-187: Please rephrase.

L190: Single or Singular?

L193: Substitute “the monthly cycle of SST” with “the monthly patterns of SST”

L190-194: The part “for data with equally spaced sampling points in time” does not fit with the rest of the sentence. Please rephrase..

L215: Substitute “followed the same trend” with “was the same”

L216: Substitute “does” with “do”, “sizes” with “size”, “or” with “and”

L217: Dolphinfish populations

Fig 1: Please add unit to x-axis

L222: Ranged between

L222: What do the numbers between the barracks mean?

Fig 2: Please use the same x-axis for all the months

L243: What do you mean with lag 0? Please explain

Fig3A: How can the prevalence be higher than 100%? Please check your data

L296: “Each third warmest”. What do you mean?

L298: Add “resulting in” after “year”. Remove “evolved”

L307-312: Speculative. What proof do you have?

L312: “The current weakens system are often described” What do you mean?

L321: Replace “occurred” with “was observed”

L322: Place “the period” behind “Although”

L327: Substitute “Infection parameters rising” with “Higher infection levels”

L327: Suitable hosts

L328: Substitute “the high temperatures” by “higher temperatures”

L328: How much evidence do you have for this? Please give some sources

L331: So do you mean these are favorable temperatures or not? Please draw a conclusion here

L338-341: More a hypothesis than a conclusion how it is formulated here

L342-355: This does still not explain the lag phase between increasing intensity and prevalence. Do fish that are already infected become more infected?

6. PLOS authors have the option to publish the peer review history of their article (what does this mean?). If published, this will include your full peer review and any attached files.

Reviewer #1: No

Reviewer #2: No

Reviewer #3: No

---

## [Author Response · Author response to Decision Letter 0]

15 Oct 2019

Thank you for your comments, which have contributed to improve the manuscript. In this new version we have supported the conclusions. 

1) We have made a generalized additive model in order to determine if the El Niño event modifies the infection parameters.

2) We use “EN” instead of “SST”, since the infection parameters changed due to a set of variables that are modified by the EN event.

3) We included new information on the egg size and total length of copepods of the non-EN and the EN samples, in order to support the results of the manuscript.

4) The paper has been professionally proofread by an English native speaker before submission. 

We hope that this new version will be good enough to be considered for publication in the journal Plos One.

---

## [Decision Letter · Decision Letter 1]

5 Dec 2019

PONE-D-19-16590R1

The 2015-2016 El Niño increased infection parameters of copepods on Eastern Tropical Pacific dolphinfish populations

PLOS ONE

Dear Dr. Santana-Piñeros,

Thank you for submitting your manuscript to PLOS ONE. After careful consideration, we still  feel that it has merit but does not fully meet PLOS ONE’s publication criteria as it currently stands. Therefore, we invite you to submit a revised version of the manuscript that addresses the points raised during the review process.

Specifically, I fully agree with Reviewer 1, concerning the need to take into consideration the limitations of the data set (short as to assess effects of interannual ENSO variability) and the inherent limitation of correlative approaches to establish causative relationships between ENSO and biological factors). This manuscript would be improved if the limitations of the approach and data set are acknowledged and alternative explanations for the results are discussed.

We would appreciate receiving your revised manuscript by Jan 19 2020 11:59PM. To enhance the reproducibility of your results, we recommend that if applicable you deposit your laboratory protocols in protocols.io, where a protocol can be assigned its own identifier (DOI) such that it can be cited independently in the future. For instructions see: http://journals.plos.org/plosone/s/submission-guidelines#loc-laboratory-protocols

We look forward to receiving your revised manuscript.

Kind regards,

Jose M. Riascos, Ph.D.

Academic Editor

PLOS ONE

Journal Requirements:

Reviewers' comments:

Reviewer's Responses to Questions

**Comments to the Author**

1. If the authors have adequately addressed your comments raised in a previous round of review and you feel that this manuscript is now acceptable for publication, you may indicate that here to bypass the “Comments to the Author” section, enter your conflict of interest statement in the “Confidential to Editor” section, and submit your "Accept" recommendation.

Reviewer #1: (No Response)

Reviewer #4: All comments have been addressed

2. Is the manuscript technically sound, and do the data support the conclusions?

Reviewer #1: Partly

Reviewer #4: Yes

3. Has the statistical analysis been performed appropriately and rigorously? 

Reviewer #1: Yes

Reviewer #4: Yes

4. Have the authors made all data underlying the findings in their manuscript fully available?

Reviewer #1: Yes

Reviewer #4: Yes

5. Is the manuscript presented in an intelligible fashion and written in standard English?

Reviewer #1: No

Reviewer #4: Yes

6. Review Comments to the Author

Reviewer #1: please see attached for my general and specific comments regarding this resubmitted manuscript to Plos One

Reviewer #4: I have carefully read the paper as well as the comment made by the previous three referees. The authors have responded all the comments made by all of them. I only have one additional comment that does not prevent the publication of this manuscript. It is related with the period analysed because, bearing in mind the complexity of the studies about influence of climatic factors, global change, etc, the period analysed could be rather short. However, as I said before, the conclusions seem to be supported by the data and, due to the lack of studies in this particular issue of ectoparasites affecting marine resources and affected by oceanographic-atmospheric factors, the manuscript is valuable.

7. PLOS authors have the option to publish the peer review history of their article (what does this mean?). If published, this will include your full peer review and any attached files.

Reviewer #1: No

Reviewer #4: No

---

## [Author Response · Author response to Decision Letter 1]

18 Feb 2020

General comments:

This remains an interesting manuscript connecting a regional warming event to parasite loads in an important fishery species, making projections regarding effects of long-term climate change on the same. However, it also remains fraught with errors in English spelling and grammar, and fails to consider alternative hypotheses or the limits of the correlative evidence presented for mechanistic inference. For example, can you explore other potential hypotheses regarding the mechanism of host-driven increases in parasite loads during the dry season, when

hosts were smaller? Maybe they are less well fed and thus weaker in responding to parasites, or they are migrating from a different water mass during the dry season – maybe even representing a different host population? If you are consistently seeing smaller dolphinfish during the dry season I find it more likely that you are seeing different fish during this period of the year than that you are seeing the same population in both seasons and that these individuals are becoming smaller and then bigger again on a yearly basis. In addition, the seasonal pattern (dry/cold vs wet/warm) is very similar to the LN-EN difference in terms of water temperature, dolphinfish size, and infection parameters. This suggests similar driving mechanisms.

A= Thank you for your comments, which have contributed to improve the manuscript. In this new version we have include other potential hypotheses for explained our result. These changes can be observed in the section “infection parameters annual patterns”. Additionally, the paper has been professionally proofread by an English native speaker before submission

I hope that this new version will be good enough to be considered for publication in the journal.

Specific comments

Line 38: modified by El Niño, affecting several ecological processes. Parasites and other marine organisms respond to environmental variation, but the influence…

A = Done

Line 67: (and throughout) During El Niño (EN) events (no “the” before El or ENSO)

A = Done

Line 75: performance

A = Done

Line 121: both temperature ranges should be in low-high order, and you cannot cite a website directly in manuscript text (please see Plos One literature citation guidelines)

A= Done

Line 194: should this be a two-way ANOVA with season and EN as crossed or nested factors?

A = Changed “one” by “two”

Line 246: different (not difference)

A= Done

Line 333: This is the first tropical survey to provide evidence that the EN period promotes…

A = Done

Line 342: widely (not profusely)

A = Done

Line 348: Sentence needs to be re-written, whole document checked for basic English

A = We re-written sentence. Manuscript has been revised by an English native speaker

Line 351: although it prefers an SST range of 21-30 C

A= Done

Line 360: Sentence needs to be re-written

A= We re-written sentence

Line 375: missing a period

A= This sentence was eraser

Line 376: an historical event

A= Done

Line 385: My experience seeing fish in the water has been that larger animals tend to have far more ectoparasites than smaller individuals.

A= We explained this result in line 383 to 395

Line 416: Sentence needs to be re-written

A= We re-written sentence

Line 420: temperature on the copepod reproductive parameters of copepods; also this sentence is a run-on, with changes in tense that need to be corrected

A= We corrected this sentence

Line 454: Apparently; more frequent in the ??? no citation? And next line cites a website? Too many formatting, stylistic, and grammatical errors. 

A= We corrected this sentence

---

## [Decision Letter · Decision Letter 2]

18 Mar 2020

PONE-D-19-16590R2

The 2015-2016 El Niño increased infection parameters of copepods on Eastern Tropical Pacific dolphinfish populations

PLOS ONE

Dear Dr. Santana-Piñeros,

Thank you for submitting your manuscript to PLOS ONE. After careful consideration, we feel that it has merit but does not fully meet PLOS ONE’s publication criteria as it currently stands. Therefore, we invite you to submit a revised version of the manuscript that addresses the points raised during the review process.

We would be willing to accept your manuscript, provided that you follow/provide convincing explanations to each suggestion or comment raised by the reviewer.

We would appreciate receiving your revised manuscript by May 02 2020 11:59PM. To enhance the reproducibility of your results, we recommend that if applicable you deposit your laboratory protocols in protocols.io, where a protocol can be assigned its own identifier (DOI) such that it can be cited independently in the future. For instructions see: http://journals.plos.org/plosone/s/submission-guidelines#loc-laboratory-protocols

We look forward to receiving your revised manuscript.

Kind regards,

Jose M. Riascos, Ph.D.

Academic Editor

PLOS ONE

Reviewers' comments:

Reviewer's Responses to Questions

**Comments to the Author**

1. If the authors have adequately addressed your comments raised in a previous round of review and you feel that this manuscript is now acceptable for publication, you may indicate that here to bypass the “Comments to the Author” section, enter your conflict of interest statement in the “Confidential to Editor” section, and submit your "Accept" recommendation.

Reviewer #1: (No Response)

2. Is the manuscript technically sound, and do the data support the conclusions?

Reviewer #1: Partly

3. Has the statistical analysis been performed appropriately and rigorously? 

Reviewer #1: Yes

4. Have the authors made all data underlying the findings in their manuscript fully available?

Reviewer #1: Yes

5. Is the manuscript presented in an intelligible fashion and written in standard English?

Reviewer #1: Yes

6. Review Comments to the Author

Reviewer #1: Please see the uploaded file for reviewer comments to the author for changes requested to MS text and methodology

7. PLOS authors have the option to publish the peer review history of their article (what does this mean?). If published, this will include your full peer review and any attached files.

Reviewer #1: No

---

## [Author Response · Author response to Decision Letter 2]

17 Apr 2020

Reply to referees’ comments. Each query is followed by an answer (A).

General comments:

Santana-Piñeros et al. have submitted their revised version of a study investigating the effects of the 2015-16 El Niño event on ectoparasite loads of dolphinfish. The manuscript has been thoroughly revised and most of the grammatical/English issues have been corrected (a few minor changes still needed, see specific comments). In addition, the authors have modified their language to consider a more nuanced explanation of the environmental factors driving variation in ectoparasite infection rates, in light of the relatively short sampling window around the major EN event.

For the most part I feel that our issues raised have been dealt with. However, there are still some inconsistencies/shortcomings in the interpretation of the results. For instance, the bulk of the article (including the title!) suggest that EN increased infection parameters of copepods on dolphinfish, yet in the discussion (see lines 420-421), you state that infection rates of C. bonito were actually higher during the non-EN period. 

A= Thank you for your comments, which have substantiallyt improved the manuscript.

We have reviewed the statiscal analyzes presented in the previous version of the manuscript and detected a mistake in the SST values, which were for the 3.4 region and not 1+2 region. We have corrected it and the generalized models showed a higher explained deviance and even the prevalence of C. quaternia could be explained. Our results confirmed that the increase in SST has risen the infection parameters of the two parasite copepods (see Figs. 3D, 4D). In this new version, we re-written the paragraph (lines 427-439) and we have included a figure in a supplementary file. 

In addition, you state that mean host TL did not differ between EN and non-EN periods, but then in the discussion (e.g., line 427) you use increased TL during EN as an explanation for decrease in C. bonito infection during EN. If you truly think that host size was an important factor interacting with the oceanographic variables to predict infection rates, you might consider analyzing different size classes separately.

A= This argument was removed from the manuscript. We separately analyzed different size classes and there were no differences in prevalence with respect to the previous analyzes. 

Finally, I have some doubts about the spectral analysis methodology. Since this is a relatively short window of time (24 months), a peak spectral density of 11 months is nothing more than saying that one year was different from the next. This alludes to my original concern with this manuscript, that the short sampling period is insufficient to establish an EN effect per se, but rather a difference between years that might be explained by oceanographic parameters. For this reason I wonder if ONI is really an informative explanatory variable or if you should simply focus on SST, salinity, etc.

A= Spectral analysis methodology does not allow comparison between years, but repeated patterns over time. The longer the time series, the different time patterns can be detected. Due to our short time series, the spectral analysis detected that every 11 months the environmental variability in the study area is repeated, which is consistent with the oceanographic temporality in that area. The difference between years was given by the SST, with higher values in 2015 (Fig. 2). In this sense, we rewrote the paragraph taking the reviewers' suggestion, focusing on explaining the results with SST and not ONI.

Specific comments

Line 39: the influence of the EN cycle… (“the” is placed before EN in this case)

A= Done.

Line 44: since the term “TEP” does not appear later in the abstract, no need to define acronym here (just say Tropical Eastern Pacific). Acronym is correctly defined in the main body of the text on line 65.

A= Done.

Line 51: Consider reporting specific number of degrees C SST increase here (rather than just “SST increased consistently”)

A= We add information in abstract (Lines 48-56) and include a specific value of SST (Line 57).

Line 52: “The SST increase interrupted the seasonal patterns of salinity and Chl-a. It is suggested that EN had the greatest influence on the infection parameters of both copepod species.” -- These two sentences do not make sense and do not add information to the abstract. You state in the previous line that only SST increased over the EN period, and the entire paper is about the influence of EN on infection parameters of copepods. I recommend removing these two sentences and beginning next sentence with simply: “We observed…”

A= We deleted those lines and modified part of the abstract, showing the most important results of the manuscript. 

Line 64: Indent first sentence of each paragraph (see throughout MS), and in this case it is correct to start with “The El Niño…”

A= Done.

Line 74: “Affects performance, population dynamics…”

A= Done. 

Line 100: Risso, 1810 is placed in parentheses while other citations of authorities who named species were not in parentheses; please be consistent

A= International Code of Zoological Nomenclature (ICZN) establishes: In citing the name of an author, the surname is given in full, not abbreviated. The date of publication in which the name was established is added for example Mugil curema Valenciennes, 1836. However, if species was described with other genera, the author and year are set in parentheses. For example, Katsuwonus pelamis (Linnaeus, 1758) was described as Scomber pelamis Linnaeus, 1758. Since the taxonomic status of A. rochei changed, then it is mandatory to have the surname in parenthesis.

Line 119: SST was defined earlier, can simply use acronym throughout manuscript

A= Done. 

Line 135: … sampling of 154 dolphinfish to establish a baseline prevalence of infection during the non-EN period

A= Done.

Line 156: using specific literature taxonomic keys

A= Done.

Line 170: the EN conditions

A= Done.

Line 174: Values of the SST

A= Done.

Line 238: (mean 72.88 ± 12.45 SD) this is to define parameters upon first usage, can leave following values as numbers without units

A= Done.

Line 271: Please define UPS acronym upon first usage (for salinity)

A= Done.

Line 272: Please state which season and/or EN/Non-EN periods had greater/lower values for each of the environmental parameters measured (rather than just stating that they were significantly different)

A= We changed paragraph. 

Line 291: 29% - 100%

A= Done.

Line 330: altering both environmental physical and biological oceanography

A= Done.

Line 341: Results (mean? TL in dry and rainy seasons) should be presented in the Results section

A= We delete this phrase.

Line 345: Can you discuss the approximate ages associated with these sizes? Are you suggesting these fish are being born and growing to maturity within a single season? Or are larger individuals migrating into these waters during the rainy season?

A= We cannot associate size classes with ages. We re-wrote this paragraph. 

Line 358: I have a hard time interpreting the 11-month cycles statement. With only 24 months of data, you can really only say that one year was different from the next.

A= The time series detects patterns over time. In this sense, we do not say that one year is different from other, but that the variability of our oceanographic variables is repeated in 11-month cycles. These cycles are consistent with the dynamics of currents and seasonal temporality in the area.

Line 360: varies varied

A= Done.

Line 398: hosts

A= Done.

Line 407: dynamics of the parasites

A= Done.

Line 418: these are different mean SST values reported as long-term averages for these months than those reported on line 120

A= We change SST by air temperature in line 120.

Line 420-421: This statement runs directly in opposition to the title of the paper

A= We modified this statement.

Line 465: density-dependence, intra- and inter-specific competition,

A= Done.

Line 469: The future ENSO events

A= Done.

---

## [Editor Report · Decision Letter 3]

22 Apr 2020

The 2015-2016 El Niño increased infection parameters of copepods on Eastern Tropical Pacific dolphinfish populations

PONE-D-19-16590R3

Dear Dr. Santana-Piñeros,

We are pleased to inform you that your manuscript has been judged scientifically suitable for publication and will be formally accepted for publication once it complies with all outstanding technical requirements.

With kind regards,

Jose M. Riascos, Ph.D.

Academic Editor

PLOS ONE
---

## [Editor Report · Acceptance letter]

27 Apr 2020

PONE-D-19-16590R3 

The 2015-2016 El Niño increased infection parameters of copepods on Eastern Tropical Pacific dolphinfish populations 

Dear Dr. Santana-Piñeros:

I am pleased to inform you that your manuscript has been deemed suitable for publication in PLOS ONE. Congratulations! Your manuscript is now with our production department. 

With kind regards,

on behalf of

Professor Jose M. Riascos 

Academic Editor

PLOS ONE